# Fast and Sensitive Bacteria Detection by Boronic Acid Modified Fluorescent Dendrimer

**DOI:** 10.3390/s21093115

**Published:** 2021-04-30

**Authors:** Ayame Mikagi, Riho Tsurufusa, Yuji Tsuchido, Takeshi Hashimoto, Takashi Hayashita

**Affiliations:** 1Department of Materials and Life Sciences, Faculty of Science and Technology, Sophia University, 7-1 Kioi-cho, Chiyoda-ku, Tokyo 102-8554, Japan; iris.14mls@eagle.sophia.ac.jp (A.M.); r-tsuruhusa-r43@eagle.sophia.ac.jp (R.T.); y-tsuchido@aoni.waseda.jp (Y.T.); t-hasimo@sophia.ac.jp (T.H.); 2Department of Life Science and Medical Bioscience, School of Advanced Science and Engineering, Waseda University (TWIns), 2-2 Wakamatsu-cho, Shinjuku-ku, Tokyo 162-8480, Japan

**Keywords:** bacteria detection, *E. coli*, *S. aureus*, dendrimer, phenylboronic acid, fluorescence measurement

## Abstract

This study reports a novel, fast, easy, and sensitive detection method for bacteria which is urgently needed to diagnose infections in their early stages. Our work presents a complex of poly(amidoamine) dendrimer modified by phenylboronic acid and labeled by a fluorescent dansyl group (Dan-B8.5-PAMAM). Our system detects bacteria in 20 min with a sensitivity of approximately 10^4^ colony-forming units (CFU)·mL^−1^. Moreover, it does not require any peculiar technical skills or expensive materials. The driving force for bacteria recognition is the binding between terminal phenylboronic acids on the probe and bacteria’s surface glycolipids, rather than electrostatic interactions. The aggregation caused by such binding reduces fluorescence. Even though our recognition method does not distinguish between live or dead bacteria, it shows selective antibacterial activity towards Gram-negative bacteria. This study may potentially contribute a new method for the convenient detection and killing of bacteria.

## 1. Introduction

In recent years, development of multidrug-resistant bacteria is becoming a global issue [1,2]. Since the abuse of antibiotics is a major reason for the genetic mutation, more specific dosing is urgently needed [3,4]. In order to detect and diagnose infections in their early stages, species-specific recognition of bacteria is particularly important. However, currently, a major recognition system for bacteria would need expensive reagents [5,6] or traditional cultivation for several days [7]. Unfortunately, to the best of our knowledge, a fast, easy, and sensitive detection method for bacteria has not been established yet in clinical practice.

The problem has attracted the interest of many researchers, and various nanomaterials conjugated with recognition elements which can recognize bacteria [8,9] or microbial agents [10] have been reported. For instance, matrix-assisted laser desorption ionization time-of-flight mass spectrometry (MALDI-TOF-MS) [11] and the disposable colorimetric sensor array [12] shortened the identification time to a few hours. An immunosensor [13] or polymerase chain reaction (PCR) [14] is also a sensitive and specific detection method for bacteria, but it requires expensive instruments and careful sample preparation, and it takes several hours to provide results. As another example, though an antibody-based system such as the enzyme-linked immunosorbent assay (ELISA) does not need a skilled researcher or a long period of time, it costs money because it uses expensive antibodies [7,15]. In the same manner as many biological challenges [16,17], nanoprobes with binding sites as a recognitive function represent a potential solution. For instance, nanoprobes with dipicolylamine for *Staphylococcus aureus* [18] or *Escherichia coli* [19,20], mannose towards *E. coli* [21,22], cationic amine groups [23,24] and vancomycin [25] for various species, or cell wall binding domains for corresponding bacteria [26] were developed to attempt instant and sensitive detection of bacteria. Many laboratories have focused on bacteria detection by convenient optical measurements, such as luminescence changing [27,28]. From the perspective of surface glycolipids on bacteria which show species specificity, the detection of saccharides might be a significant strategy. Phenylboronic acid is widely applied for saccharide recognition because it is known that phenylboronic acid forms a bond with the *cis*-diol sites of saccharide [29,30]. Previously, we reported the selective recognition of Gram-positive bacteria by a poly(amidoamine) dendrimer chemically modified by phenylboronic acid (B-PAMAM) [31]. Phenylboronic acids of B-PAMAM interacted with the surfaces of bacteria and formed aggregations which could be observed by the naked eye. However, the limit of detection (LOD) calculated from turbidity measurements was 6.4 × 10^6^ CFU·mL^−1^. This value was far from the clinical requirement, which is approximately 10^3^ or 10^5^ CFU·mL^−1^ [32,33]. Moreover, the biological activity of B-PAMAM and the status of bacteria (live or dead) have not been investigated at all. This study aims to improve the sensitivity of fluorescence measurement and investigate the antibacterial activity of dendrimers. A dansyl group (Dan) was introduced to B-PAMAM as a fluorescence label [34,35] and different kinds of dendrimers were prepared depending on the number of phenylboronic acid groups attached on their surfaces (Figure 1). Fluorescent measurements carried out by Dan-B-PAMAMs proved to be fast, easy, and sensitive.

## 2. Materials and Methods

### 2.1. Reagents and Apparatus

#### 2.1.1. Reagents

All reagents and solvents were obtained from commercial suppliers and used without further purification, unless otherwise stated. Firstly, 4-Carboxyphenylboronic acid (354-36403), triethylamine (202-02641), sodium chloride (191-01665), and agar (018-15811) were purchased from FUJIFILM Wako Pure Chemical Corporation (Osaka, Japan). Dimethylsulfoxide luminasol (DMSO-Lu, LU08), propidium iodide solution (PI, P378), 3,6-bis(dimethylamino)acridine hydrochloride solution (AO, A430), ethidium bromide solution (EB, E272), 3-(4,5-dimethyl-2-thiazolyl)-2,5-diphenyl-2*H*-tetrazolium bromide (MTT, M009), and 4′,6-diamidino-2-phenylindole dihydrochloride n-hydrate (DAPI, D523) were purchased from DOJINDO LABORATORIES (Kumamoto, Japan). Moreover, 5-(Dimethylamino)naphthalene-1-sulfonyl chloride (dansyl chloride, D0656) and 4-(4,6-dimethoxy1,3,5-triazin-2-yl)-4-methylmorpholinium chloride (DMT-MM, D2919) were purchased from Tokyo Chemical Industry Co. Ltd. (Tokyo, Japan). Methanol (25183-70), methanol-d_4_ (99.8%, 25183-70), and D_2_O (99.8%, 10086-23) were purchased from Kanto Chemical Co. Inc. (Tokyo, Japan). Poly(amidoamine) (PAMAM) dendrimer, ethylenediamine core, generation 3.5 (142430-2.5G) and 4.0 (412449-10G) solutions were purchased from Sigma-Aldrich Japan Co. LLC. (Tokyo, Japan). Bacto yeast extract (212750) and bacto tryptone (211705) were purchased from Nippon Becton, Dickinson Co., Ltd. (Tokyo, Japan). Phosphate-buffered saline (PBS, 2101) was purchased from Cell Science & Technology Institute Inc. (Miyagi, Japan). Water was doubly distilled and deionized using a Milli-Q water system (WG222, Yamato Scientific Co., Ltd., Tokyo, Japan and Milli-Q Advantage, Merck Millipore, Burlington, MA, USA) before use.

#### 2.1.2. Apparatus

^1^H Nuclear magnetic resonance (NMR) spectra were recorded on a JEOL JNM-ECA 500 spectrometer (500 MHz) by JEOL (Tokyo, Japan) at 300 K or a Bruker Avance III HD 400 MHz at 298 K in deuterated solvents. All pH values were recorded using a Horiba F-52 pH meter (HORIBA Ltd., Kyoto, Japan). Ultraviolet−visible (UV–vis) absorption spectra were measured using a Hitachi V-570 or V-760 UV–vis spectrophotometer (Hitachi High-Technologies Co., Tokyo, Japan) equipped with a Peltier thermocontroller and a 10 mm quartz cell. Fluorescence spectra were measured using a HITACHI F-7000 fluorescence spectrophotometer (Hitachi High-Technologies Co., Tokyo, Japan) equipped with a Peltier thermocontroller and a 10 mm quartz cell. Zeta potential measurements were carried out at 25 °C using a Zetasizer Nano ZS (Malvern Instruments Ltd., Malvern, Worcestershire, UK). Samples were shaken by MALTI CHAKER MS-300 (AS ONE Co., Osaka, Japan). Centrifugation was conducted using CF15RN (Hitachi High-Technologies Co., Tokyo, Japan). Nanoprobes were lyophilized by EYELA FDU-1200 (TOKYO RIKAKIKAI Co. LTD., Tokyo, Japan).

### 2.2. Preparation of Dendrimer Probes

#### 2.2.1. Preparation of PAMAM(NH_2_)

PAMAM(G4) dendrimer dissolved in methanol (1.0 mL, 5.7 μmol) was transferred into a Spectra/Por 6 dialysis bag (MW cutoff = 1000, Repligen, MA, USA) and dialyzed against distilled water. The solution was lyophilized for 2 days to produce a white powder (63.7 mg, 78%).

#### 2.2.2. Preparation of PAMAM(COOH)

PAMAM(G3.5) dendrimer dissolved in methanol (1.5 mL, 8.6 μmol) was transferred into a Spectra/Por 6 dialysis bag (MW cutoff = 1000, Repligen, MA, USA) and dialyzed against distilled water. The solution was lyophilized for 2 days to give a white powder (84.9 mg, 70%).

#### 2.2.3. Synthesis of Dan-PAMAM

PAMAM(G4) dendrimer dissolved in methanol (4.5 mL, 25.7 μmol), 5-(dimethylamino)naphthalene-1-sulfonyl chloride (41.39 mg, 153 μmol), and triethylamine (1.172 g, 11.6 mmol) was dissolved in 30 mL of methanol followed by reflux at 60 °C for 72 h. The mixture was transferred into a Spectra/Por 6 dialysis bag (MW cutoff = 1000, Repligen, MA, USA) and dialyzed against methanol and distilled water. The resultant solution was lyophilized to obtain a yellow solid (288 mg, 72.9%). The structure and the number of dansyl substitutions were confirmed by ^1^H NMR spectra (Appendix A).

#### 2.2.4. Synthesis of Dan-B-PAMAMs

Dan-PAMAM, 4-carboxyphenylboronic acid, and 4-(4,6-dimethoxy-1,3,5-triazin-2-yl)-4-methylmorpholinium chloride were dissolved in methanol (20 mL) and stirred at room temperature for 2 days (Table 1). The reaction mixture was transferred into a Spectra/Por 6 dialysis bag (MW cutoff = 1000, Repligen, MA, USA) and dialyzed against methanol and distilled water. The solution was then lyophilized to give a pale-yellow solid, followed by ^1^H NMR measurement. The number of phenylboronic acid substitutions was calculated from the peak area of ^1^H NMR spectra (Appendix A). It was noted that Dan-B3-PAMAM was obtained from Dan-PAMAM which was modified by six dansyl groups (Appendix A).

### 2.3. Bacteria Detection

#### 2.3.1. Culturing of Bacteria

Lysogeny broth (LB) medium was prepared using 2 g of bacto tripton, 1 g of bacto yeast extract, and 2 g of NaCl in 200 mL distilled water. *S. aureus* IAM1011 and *E. coli* K12W3110 were cultured at 37 °C overnight on LB agar plates which contained mixtures of 200 mL of LB medium and 3 g of agar. The obtained colony was isolated and cultured in LB medium at 37 °C overnight. The solution was centrifuged (10,000 rpm, 1 min) and washed with distilled water. The procedure was repeated twice to remove the medium. After washing, it was centrifuged once more and washed with PBS buffer. The concentrations of the bacterial suspensions were adjusted by measuring OD_600_ after vortex mixing. Calculation index is shown below.

*S. aureus*: OD_600_ = 1.0, CFU = 4.5 × 10^8^ mL^−1^*E. coli*: OD_600_ = 1.0, CFU = 1.0 × 10^9^ mL^−1^

Cultured bacteria were subjected to fluorescence microscopy imaging, optical measurements, and zeta potential measurements. For fluorescence microscopy, DAPI, EB, AO or PI solution was mixed to cultured bacteria in PBS buffer and washed to displace the dye. Fluorescence intensity and zeta potential were obtained at room temperature, while UV–vis spectra and OD_600_ were measured at 25 °C.

#### 2.3.2. Detection Method without Filtration

Dan-B3-PAMAM (1.5 mL, 6.6 μM) and *S. aureus* in PBS buffer (1.5 mL, 2.0 × 10^5^ to 2.0 × 10^8^ CFU·mL^−1^) were mixed for 10 min at 2000 rpm followed by fluorescence and turbidity measurements with or without 10 min standing.

#### 2.3.3. Optimized Detection Method with Filtration

Dan-PAMAM, Dan-B4-PAMAM, or Dan-B8.5-PAMAM (1.5 mL, 6.6 μM) and *S. aureus* (1.5 mL, 2.0 × 10^4^ to 2.0 × 10^9^ CFU·mL^−1^) or *E. coli* (1.5 mL, 2.0 × 10^4^ to 2.0 × 10^8^ CFU·mL^−1^) in PBS buffer were mixed for 10 min at 2000 rpm. The resultant solution was left to stand for 10 min and filtered (the diameter of pores was 0.2 μm), followed by fluorescence measurement. Then, 1.5 mL of PBS buffer was selected as a bacterial control sample and subjected to the same procedure. Reproducibility was confirmed by repeating three or more times. 

#### 2.3.4. MTT Assay

Bacteria (750 μL, 6.0 × 10^8^ CFU·mL^−1^) and probe solution (750 μL, 6.6 μM) or PBS buffer (750 μL) were mixed and shaken for 10 min, 2000 rpm to form complex. After the centrifugation of the solution, the supernatant was removed from complex. LB medium (1 mL) and MTT solution (100 μL, 5 mg·mL^−1^) were mixed with the complex, followed by incubation at 37 °C for 20 min [36]. After incubation, the reaction mixture was centrifugated and supernatant was displaced from the precipitates. The resultant solution was added to 1 mL of DMSO-Lu. DMSO-Lu destroyed the bacteria’s membrane and quenched the reaction of MTT by mixing. The absorbance of dissolved formazan was measured (*λ*_abs_ = 560 nm) and PBS samples were used as controls.

## 3. Results and Discussion

### 3.1. Recognition Mechanism by Chemically Modified PAMAM Dendrimers

The surface properties of various PAMAM dendrimers were investigated by zeta potential measurement to give information about the interaction with bacteria [37]. For each sample, the measurements were repeated three times for PAMAM(NH_2_) and PAMAM(COOH) and six times for other PAMAMs with terminal dansyl groups. The average values are summarized in Table 2.

The results of PAMAM(NH_2_) and PAMAM(COOH) showed that the amine group of PAMAM(NH_2_) was protonated and carboxylic acid on PAMAM(COOH) was deprotonated at pH 7.4. The measurements of Dan-PAMAM, Dan-B4-PAMAM, and Dan-B8.5-PAMAM showed a positively charged surface. The value of their zeta potential decreased as the substituent number of phenylboronic acids increased. Such results suggested that phenylboronic acid forms a tetrahedral boronate at pH 7.4, and it causes the neutralization of amine’s positive charges, weakening their electrostatic interactions. Since it is well known that the surface of bacteria is negatively charged [38], positively charged probes could form minute aggregates by electrostatic interaction with bacteria. Fluorescent microscope observations clearly confirmed this hypothesis (Figure 2A–D).

The microscopy images showed that the solution of PBS buffer as a control sample and PAMAM(COOH) did not produce any aggregates, in contrast to other probes (Figure 2A,B). Among probes modified by the dansyl group, images of dansyl and EB describing the existence of bacteria were overlapped (Figure 2C,D). The results indicated that probes were able to form aggregates with bacteria. Even though PAMAM(NH_2_) and Dan-PAMAM had not been functionalized with phenylboronic acid groups, they formed minute aggregates with both *S. aureus* and *E. coli*. Such aggregations suggested that electrostatic bonding between positively charged PAMAM(NH_2_) or Dan-PAMAM and negatively charged bacteria was sufficient for the success of their interactions, and the implementation of phenylboronic acids was not necessary for forming minute aggregates. In terms of Dan-B4-PAMAM and Dan-B8.5-PAMAM, aggregations were observed as well and Dan-B8.5-PAMAM gave much larger aggregates than Dan-B4-PAMAM. The difference can be explained as follows. The bonding between terminal phenylboronic acids and glycolipids on the bacteria surface is the critical driving force of the aggregation, rather than electrostatic interaction. It was suggested that the more phenylboronic acid groups bonded on dendrimers, the more bacteria interacted and formed large aggregates.

### 3.2. Development of Sensitive Bacterial Recognition

Dan-B3-PAMAM was used as a test probe to develop a novel sensitive method using fluorescence measurements (Figure 3A,B). In this study, fluorescent aggregates between probes and bacteria were observed after being mixed for only 10 min. The transformation of fluorescence intensity after the formation of aggregates is a desirable index. Fluorescence and OD_600_ measurements of supernatants were carried out twice: after mixing and after standing for 10 min [31].

Unfortunately, fluorescence intensity did not decrease significantly in all samples (Figure 3A). Only the turbidity of the 10^8^ CFU·mL^−1^ concentration changed significantly after standing for 10 min (Figure 3B). Minute aggregations and bacteria floating in the sample solution caused light scattering and affected the measurements. Therefore, a protocol for deleting floating particles is deemed to be necessary to improve sensitivity. Based on this hypothesis, minute compounds were removed by filtration (the diameter of pores was 0.2 μm) after standing for 10 min, and fluorescent measurements were conducted only for the filtrate solution (Figure 4A,B). A mixture of various probes and PBS buffer was prepared as a negative control (Figure 4A).

Measurements carried out after filtration showed good reproducibility (Figure 4A) and the reduction of fluorescent intensity too (Figure 4B). In particular, the Dan-B8.5-PAMAM sample produced visible aggregates and a notable reduction in terms of intensity change. The reduction of fluorescence was also confirmed in the Dan-PAMAM solution, which should not recognize bacteria. The change could be explained by the elimination of minute aggregations formed by electrostatic interactions. According to the result of the Dan-B4-PAMAM sample, it was suggested that the recognition power was due to the number of phenylboronic acids of dendrimers able to bind with saccharide sites on the bacteria surface.

Afterwards, the recognition capability of Dan-B8.5-PAMAM and the negative control, Dan-PAMAM, was evaluated (Figure 5A,B). The maximum bacterial concentration was confirmed as a concentration in which aggregates could no longer be seen by the naked eye (*S. aureus*: 10^7^ CFU·mL^−1^, *E. coli*: 10^9^ CFU·mL^−1^).

Only Dan-B8.5-PAMAM produced a significant reduction in fluorescence with both bacteria as expected (Appendix A). Even in low concentrations of bacteria, Dan-B8.5-PAMAM successfully showed consistent performance with a small margin of error. Considering that Dan-PAMAM did not show significant changes in 10^4^–10^5^ CFU·mL^−1^ of bacteria, phenylboronic acid groups proved to be the origin of such sensitive recognition. Among high-concentration samples of *E. coli*, Dan-PAMAM demonstrated a significant reduction in fluorescence (Figure 5B). This could be explained by the Dan-PAMAM electrostatic interactions that formed minute aggregates. LOD was not calculated since linearity was not obtained for low concentrations. It was suggested that negligible operational errors affected the dispersion results, since the decrease in fluorescence was minor. Even though selectivity was not observed, this filtration protocol significantly improved the sensitivity of the turbidity method [31] (approximately 10^3^ times). It should be noted that the novel method could detect 10^4^ CFU·mL^−1^ of bacteria and it met this requirement in some clinical settings [32,33].

### 3.3. Viability of Bacteria

In order to elucidate bacterial viability after the recognition procedure, *S. aureus* (which showed large aggregations, Figure 2C) was labeled by fluorescent compounds PI and AO. PI emits fluorescence when the bacteria surface is deconstructed, while AO shows both states of life and death. After the labeling, fluorescent microscopy images were obtained (Figure 6A).

As the images clearly demonstrate, the fluorescent sites of AO, PI, and the dansyl group overlapped in both probes. This means that aggregates with damaged bacteria appeared independently of the concentration of phenylboronic acids. However, the existence of living bacteria was not confirmed because the areas of AO and PI overlapped. Hence, an MTT assay, which could provide information about bacteria’s living ratio in a solution, was conducted (Figure 6B) [36]. In this assay, MTT was subjected to enzyme synthesis and formazan, with an absorbance at 560 nm being obtained. The absorbance of the control PBS sample was set as 1.0 and the absorbance ratio was calculated to give information about cell viability. When *S. aureus* was subjected to such an assay, almost all the bacteria were viable after the recognition protocol. Instead, over half of *E. coli* were dead in the case of Dan-B8.5-PAMAM. Since the fluorescence microscopy images depicted the existence of damaged bacteria, the Dan-B-PAMAMs might interact with bacteria independently of bacterial status. Considering the results of PAMAM(NH_2_) and Dan-PAMAM, bacterial selectivity was not observed due to amine groups or dansyl groups but from phenylboronic acid recognition sites. The relationship among the concentration of phenylboronic acids and the size of aggregates might be important because Dan-B4-PAMAM, with small aggregates, did not show any antibacterial activity. We hypothesized that, in the case of Dan-B8.5-PAMAM, *E. coli* suffered more because of the high density caused by large aggregates, and this caused the death of bacteria after recognition.

## 4. Conclusions

This report describes a fast, easy, and sensitive bacterial detection method which only takes 20 min. Fluorescence-labeled PAMAM dendrimers with phenylboronic acid modification (Dan-B-PAMAMs) were designed and their interactions with bacteria were investigated. Especially in the case of Dan-B8.5-PAMAM fluorescence measurements, visible aggregates were obtained, and the sensitivity was significantly improved to 10^4^ CFU·mL^−1^. The size of aggregates was dependent on the number of terminal phenylboronic acids on dendrimers, deemed to be the determinant for the increase in sensitivity. The major driving force of aggregation was provided by the interaction between phenylboronic acids and glycolipids situated on the surfaces of bacteria, rather than electrostatic interactions by the amino groups of dendrimer. Phenylboronic acids affected the viability of *E. coli* selectively, after its recognition. This report has provided new insights into the role of multipoint recognition which causes bacteria aggregation. Our results suggest that the size of bacteria aggregation is important, not only for recognition sensitivity but also for antibacterial activity. Even though the antimicrobial spectrum needs further investigation, this study may potentially be established as a method which provides convenient and sensitive detection and killing of bacteria.

## Figures and Tables

**Figure 1 sensors-21-03115-f001:**
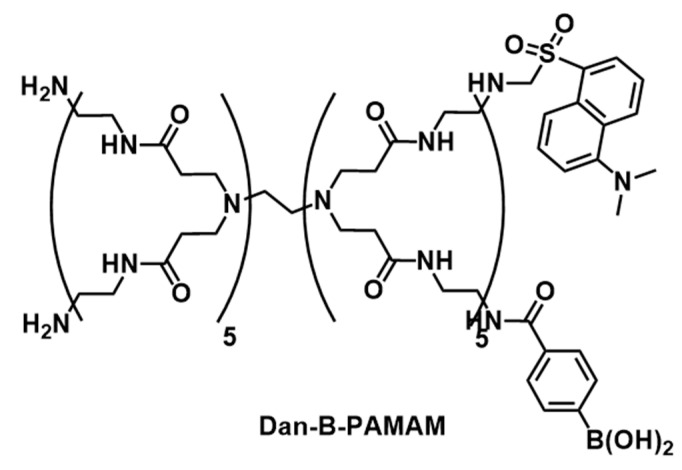
Structure of Dan-B-PAMAM.

**Figure 2 sensors-21-03115-f002:**
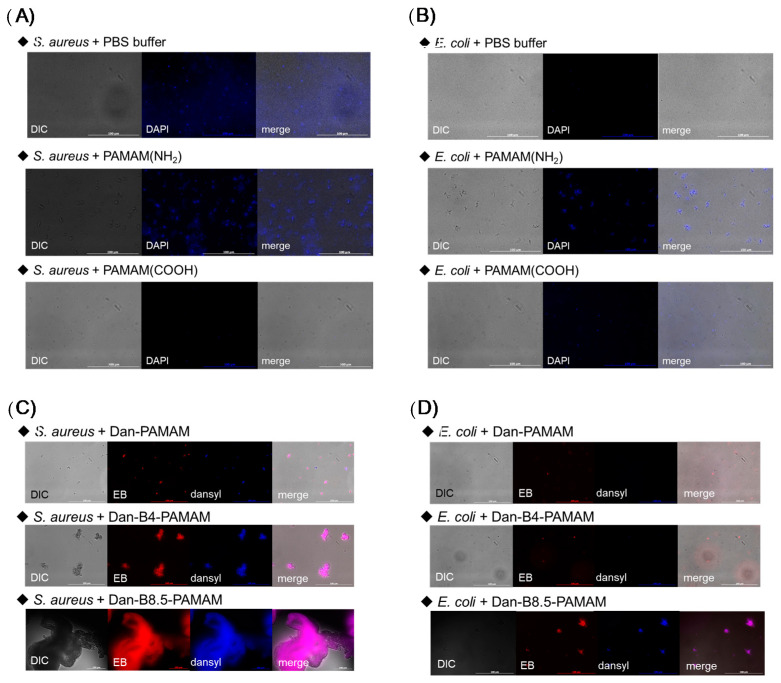
Fluorescence microscopy images of *S. aureus* or *E. coli* with dendrimers (pH = 7.4 adjusted by PBS buffer, (dendrimer) = 3.3 µM). (**A**,**B**) bacteria were stained with DAPI, *λ*_ex_ = 360 nm, OD_600_ of bacteria = 0.3. (**C**,**D**) bacteria were stained with EB, EB: *λ*_ex_ = 525 nm, probe: *λ*_ex_ = 330 nm. (bacteria) = 10^8^ CFU·mL^−1^.

**Figure 3 sensors-21-03115-f003:**
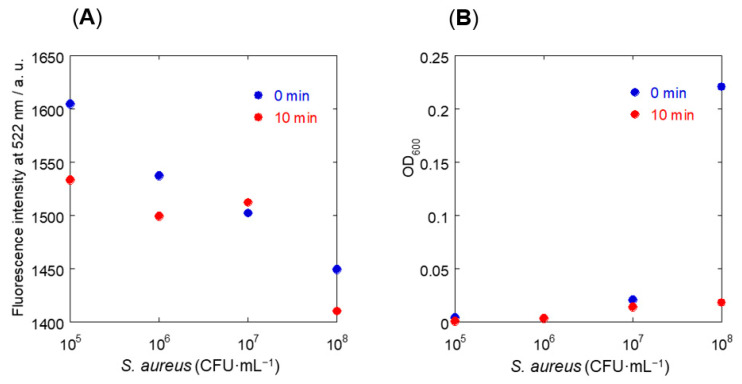
Fluorescence and turbidity measurements of Dan-B3-PAMAM in PBS buffer at pH 7.4 ((Dan-B3-PAMAM) = 0.1 mg·mL^−1^, (*S. aureus*) = 10^5^–10^8^ CFU·mL^−1^). (**A**) Fluorescence intensity at 522 nm (*λ*_ex_ = 330 nm). (**B**) OD_600_ of 0 min and 10 min.

**Figure 4 sensors-21-03115-f004:**
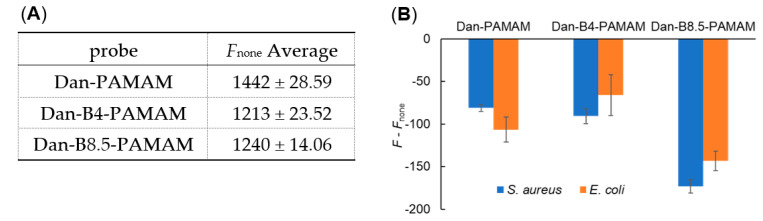
Fluorescence measurements of Dan-PAMAMs. (**A**) Average of fluorescent intensity of probes in PBS buffer adjusted at pH 7.4 (*λ*_ex_ = 330 nm, *λ*_em_ = 522 nm, (probe) = 3.3 µM, n ≥ 3). (**B**) Decrease in fluorescence intensity compared with *F*_none_ (*λ*_ex_ = 330 nm, *λ*_em_ = 522 nm), (probe) = 3.3 μM, (bacteria) = 10^8^ CFU·mL^−1^, n ≥ 3.

**Figure 5 sensors-21-03115-f005:**
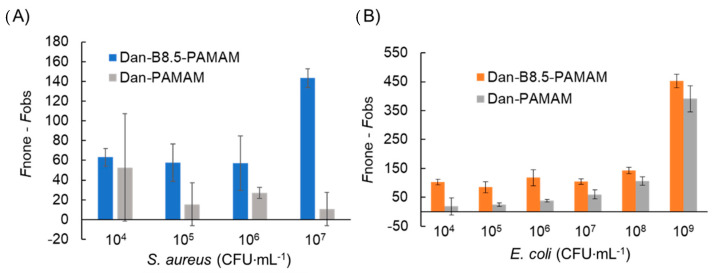
Summary of changes in fluorescence intensity (*λ*_ex_ = 330 nm, *λ*_em_ = 522 nm, (probe) = 3.3 μM, n = 3). (**A**) (*S. aureus*) = 10^4^–10^7^ CFU·mL^−1^. (**B**) (*E. coli*) = 10^4^–10^9^ CFU·mL^−1^.

**Figure 6 sensors-21-03115-f006:**
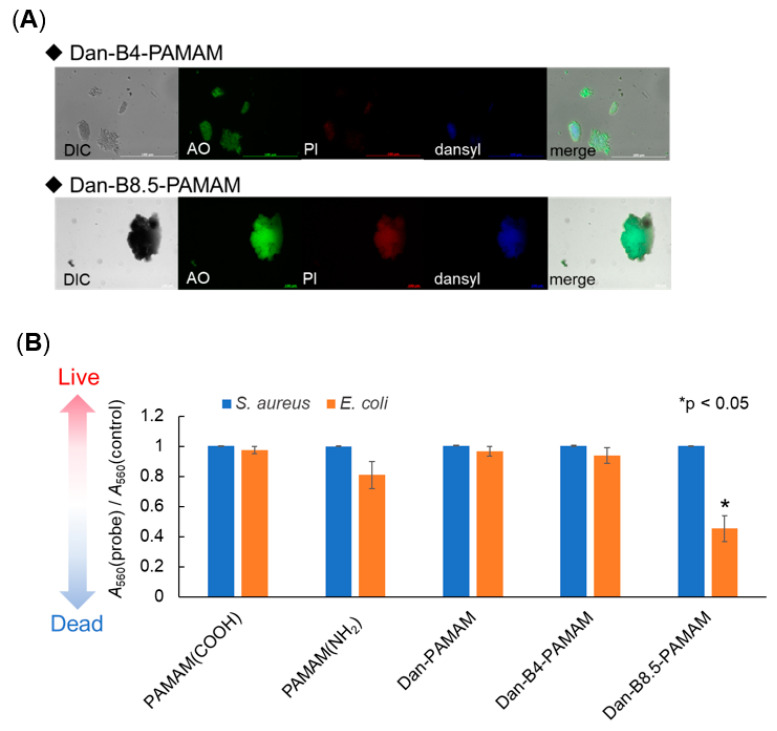
Viability of bacteria. (**A**) Fluorescence microscopy images of *S. aureus* (10^8^ CFU·mL^−1^) incubated with a probe, *S. aureus* was stained with AO and PI, AO: *λ*_ex_ = 500 nm, PI: *λ*_ex_ = 530 nm, probe: *λ*_ex_ = 330 nm. (probe) = 3.3 µM. (**B**) Absorbance change by MTT assay repeated three times for each (*λ*_abs_ = 560 nm), (probe) = 3.3 μM, (*S. aureus*) = 3.0 × 10^8^ CFU·mL^−1^, (*E. coli*) = 3.0 × 10^8^ CFU·mL^−1^. Welch’s t-test was used to compare control and probes. Differences were calculated with two-side test with an alpha level of 0.05. Asterisks indicate significant differences (*p* < 0.05).

**Table 1 sensors-21-03115-t001:** Syntheses of Dan-B-PAMAM dendrimers.

	Dan-PAMAM	Phenylboronic Acid	DMT-MM	The Number of Phenylboronic Acid Substitutions	Yield
Dan-B3-PAMAM	91.62 mg	5.84 mg (35.2 μmol)	48.71 mg (176.2 μmol)	3	31.6 mg (23%)
Dan-B4-PAMAM	43.25 mg	1.89 mg (11.4 μmol)	15.77 mg (57.03 μmol)	4	47.4 mg (86%)
Dan-B8.5-PAMAM	43.93 mg	4.71 mg (28.4 μmol)	42.99 mg (155.5 μmol)	8.5	30.2 mg (46%)

**Table 2 sensors-21-03115-t002:** Zeta potential of PAMAM dendrimers.

Probe	Zeta Potential (mV)
PAMAM(COOH)	−3.58 ± 2.08
PAMAM(NH_2_)	3.42 ± 1.54
Dan-PAMAM	13.78 ± 1.27
Dan-B4-PAMAM	6.57 ± 5.58
Dan-B8.5-PAMAM	4.30 ± 4.34

## Data Availability

Data sharing not applicable.

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
