# Peer review of "Fast and Sensitive Bacteria Detection by Boronic Acid Modified Fluorescent Dendrimer"

_sensors, 2021, doi:10.3390/s21093115_

Round 1

Reviewer 1 Report

Dear Authors,

I'm impressed with your scientific work with focus on the developing of a new method for fast detection (20 min) and killing of Gram-negative bacteria, such as opportunistic pathogen Escherichia coli, in a concentration of bacteria 104 CFU/ml. I have the following critical remarks.

  1. You claim that the PCR method is a time consuming process. This is not true and you should fix it (line 37). But so written, it seems to take days. The DNA extraction, together with the preparation of probes, PCR reaction and visualization of the PCR products on an agarose gel, usually takes about 4.30-5 hours.
  2. I recommend separating reagents from apparatus in section 2. Materials and Methods. You must add the reagent catalogue numbers. You missed to add lyophilizer information. Why did you choose lyophilisation in section 2. Preparation of Dendrimer Probes over vacuum rotary evaporation to a dry state of the sample, which is relatively faster?
  3. At the first mention of CFU/ml, LB medium (section 2.3. Bacteria Detection) please write the full name of the abbreviations. There is no information in the text about the nutrient medium used (catalog number and manufacturer). You centrifuge the bacterial inoculums 3 times at 10,000 rpm (line 126). Are you sure that the bacteria are still alive, as it is known in practice that the speed of centrifugation should not exceed 5000 rpm? The measured absorption can be misleading as it will also measure dead cells. Did you make a separate experiment with one sample, which containing only killed cells and another sample with only living cells? In this way you can mix them in different amounts and know for sure what is the percentage of living and dead cells in the sample that are detected by Dan-B8.5-PAMAM Please indicate from which collection are the test-bacterial strains. It is not necessary to write the abbreviations of bacteria in brackets after their full name (line 125). I recommend giving separate section for culturing of test-bacteria.
  4. The citation (line 147) should transferred into the text (section 2.3.3. MTT assay) and to mention only the modifications made to the protocol.
  5. Figure 2 and Figure 6 A) are too small. If possible, increase the size of images from fluorescence microscopy and improve their quality.
  6. Please correct the title of the ordinate on Figure 3 A, as it is unclear.

Please justify your answers and make corrections. I will recommend the manuscript for accepting after correction of my remarks.

Author Response

<Reply to Reviewer 4>

Thank you for the suggestive comments. The followings are responses to the comments:

1) You claim that the PCR method is a time consuming process. This is not true and you should fix it (line 37). But so written, it seems to take days. The DNA extraction, together with the preparation of probes, PCR reaction and visualization of the PCR products on an agarose gel, usually takes about 4.30-5 hours.

We sincerely appreciate this comment. The time required was revised as follows (lines 39-42); “An immunosensor [13] or polymerase chain reaction (PCR) [14] is also a sensitive and specific detection method for bacteria, but it requires expensive instruments and careful sample preparation which takes several hours for the result.”

2) I recommend separating reagents from apparatus in section 2. Materials and Methods. You must add the reagent catalogue numbers. You missed to add lyophilizer information. Why did you choose lyophilisation in section 2. Preparation of Dendrimer Probes over vacuum rotary evaporation to a dry state of the sample, which is relatively faster?

Reagents and apparatus in section 2 was separated to “2.1.2 Reagents” and “2.1.2. Apparatus” (lines 71-107).

Catalogue numbers of reagents were added as follows; (lines 73-90); “4-Carboxyphenylboronic acid (354-36403), triethylamine (202-02641), sodium chloride (191-01665), and agar (018-15811) were purchased from FUJIFILM...”

Information of lyophilizer was added (lines 106-107); “Nanoprobes were lyophilized by EYELA FDU-1200 (TOKYO RIKAKIKAI Co., LTD., Tokyo, Japan).”

When we tried vacuum by rotary evaporator, we could not obtain dry dendrimer samples completely. A rotary evaporator gave the samples as sticky oil while a lyophilizer gave them as solid or powder. Based on the results, we chose lyophilizer.

3) At the first mention of CFU/ml, LB medium (section 2.3. Bacteria Detection) please write the full name of the abbreviations. There is no information in the text about the nutrient medium used (catalog number and manufacturer). You centrifuge the bacterial inoculums 3 times at 10,000 rpm (line 126). Are you sure that the bacteria are still alive, as it is known in practice that the speed of centrifugation should not exceed 5000 rpm? The measured absorption can be misleading as it will also measure dead cells. Did you make a separate experiment with one sample, which containing only killed cells and another sample with only living cells? In this way you can mix them in different amounts and know for sure what is the percentage of living and dead cells in the sample that are detected by Dan-B8.5-PAMAM Please indicate from which collection are the test-bacterial strains. It is not necessary to write the abbreviations of bacteria in brackets after their full name (line 125). I recommend giving separate section for culturing of test-bacteria.

Full names of the abbreviations (CFU and LB) were mentioned (lines 15-16 and 140-141); “Our system detects bacteria in 20 minutes with a sensitivity about 104 colony forming unit (CFU)·mL−1.”, “Lysogeny broth (LB) medium was prepared using 2g of bacto tripton, 1 g of bacto yeast extract, and 2 g of NaCl in 200 mL distilled water.”

Information of LB medium and LB agar plate were added to the manuscript as follows (lines 140-144); “Lysogeny broth (LB) medium was prepared using 2g of bacto tripton, 1 g of bacto yeast extract, and 2 g of NaCl in 200 mL distilled water. S. aureus IAM1011 and E. coli K12W3110 were cultured at 37 °C overnight on LB agar plates which were the mixtures between 200 mL of LB medium and 3 g of agar. The obtained colony was isolated and cultured in LB medium at 37 °C overnight.” (The reagents used to prepare were written in section 2.1.2, lines 71-92.)

Even though the centrifugation and recognition protocol were carried out, fluorescence microscopy images of the mixture between S. aureus and Dan-PAMAM did not show the existence of damaged bacteria. The images and information were additionally provided in supporting information (Figure S8: Fluorescence microscopy images of S. aureus with Dan-PAMAM.).

The collections of test-bacterial strains were mentioned (line 141-142); “S. aureus IAM1011 and E. coli K12W3110 were cultured at 37 °C overnight on LB agar plates which were the mixtures between 200 mL of LB medium and 3 g of agar.”

The abbreviations of bacteria in brackets after their full name was deleted (lines 46-47) and a section of bacteria culturing was added (2.3.1. Culturing of Bacteria).

4) The citation (line 147) should transferred into the text (section 2.3.3. MTT assay) and to mention only the modifications made to the protocol.

The citation of reference 36 was moved into the text as below (lines 170-172); “LB medium (1 mL) and MTT solution (100 mL, 5 mg·mL−1) were mixed to the complex followed by incubation at 37 °C for 20 min [36].”

5) Figure 2 and Figure 6 A) are too small. If possible, increase the size of images from fluorescence microscopy and improve their quality.

Please accept our apologies for that we can no longer enhance the image resolution of Figure 2 and Figure 6 A).

6) Please correct the title of the ordinate on Figure 3 A, as it is unclear.

The title of the ordinate on Figure 3 A was changed to “Fluorescence intensity at 522 nm /a.u.”.

Reviewer 2 Report

This manuscript report about a fast and sensitive bacteria detection by boronic acid modified fluorescent dendrimer. Manuscript need following changes to improve it further.

Comments:

Line 28: Since abuse of antibiotics is a major…five references cites without providing any key information from articles. Avoid multiple citations and key information from each article discussed.

Line 34: What are various nanomaterial reported…Add more information.

Line 42: What are different bacteria detected using various materials reported here.

Line 45: it forms cis-diol bonds with saccharides…How this bonds helps in detection, need more explanation.

Section 2.3, line 126: Centrifuged three times…what you mean by this? Did you have used distilled water or any other solution for cell washing?

Improve conclusion part by adding points on future prospects of study.

Author Response

Thank you for the suggestive comments.

Lines 28, 34, 42, and 45: Key informations from each reference (including materials, methods or target bacteria) were added in the manuscript (lines 27-51).

1) Line 28: Since abuse of antibiotics is a major…five references cites without providing any key information from articles. Avoid multiple citations and key information from each article discussed.

Lines 28-32: Since abuse of antibiotics is a major reason of the genetic mutation, their much specific dosing is urgently needed [3,4]. In order to detect and diagnose infections in the early stages, species-specific recognition of bacteria is particularly important. However, up to now, a major recognition system for bacteria needs expensive reagents [5,6] or traditional cultivation for several days [7].

2) Line 34: What are various nanomaterial reported…Add more information.

Lines 35-37: The problem has attracted interests of many researchers and various nanomaterials conjugated with recognition elements which can recognize bacteria [8,9] or microbial agents [10] has been reported.

3) Line 42: What are different bacteria detected using various materials reported here.

Lines 46-51: For instance, nanoprobes with dipicolylamine for Staphylococcus aureus [18] or Escherichia coli [19,20], mannose towards E. coli [21,22], cationic amine groups [23,24] and vancomycin [25] for various species, or cell wall binding domains for corresponding bacteria [26] were developed to attempt instant and sensitive detection of bacteria. A number of laboratories focused on bacteria detection by convenient optical measurements such as luminescence changing [27,28].

4) Line 45: it forms cis-diol bonds with saccharides…How this bonds helps in detection, need more explanation.

Lines 53-55: Phenylboronic acid is widely applied for saccharides recognition because it is known that phenylboronic acid forms a bond with the cis-diol sites of saccharide [29,30].

5) Section 2.3, line 126: Centrifuged three times…what you mean by this? Did you have used distilled water or any other solution for cell washing?

Section 2.3, line 126: A detailed description was added as follows (lines 144-146); "The solution was centrifuged (10000 rpm, 1 min) and washed with distilled water. The procedure was repeated twice to remove the medium. After washing, it was centrifuged one more time and washed with PBS buffer."

6) Improve conclusion part by adding points on future prospects of study.

Conclusion part: Future prospects were added (lines 319-324); "This report has opened a new window into the role of multipoint recognition which causes bacteria aggregation. Our results suggest that the size of bacteria aggregation is important not only for recognition sensitivity but also for antibacterial activity. Even though the antimicrobial spectrum needs to be further investigated, this study may potentially be established as a method which provides convenient and sensitive detection and killing of bacteria."

Reviewer 3 Report

In this manuscript authors developed a novel, fast, and sensitive detection method for bacteria needed to diagnose infections in the early stages using a complex dendrimer consisting of phenylboronic acid as Dan-B8.5-PAMAM. Authors showed that their system detects bacteria in 20 minutes with a sensitivity about 104 CFU mL−1 showing its superior performance over their previous fluorescent probe (ref 31). The main driving force for bacteria recognition is the binding between phenylboronic acids and bacteria’s surface glycolipids, resulting decrease in fluorescence. The results look quite promising and the study may potentially contribute a new method for convenient detection and killing of bacteria. I recommend the findings for publications in Sensors.

Author Response

Thank you very much for encouraging our results for publication in Sensors.

Reviewer 4 Report

This paper describes the quick and sensitive test of bacteria detection. Fluorescently labeled PAMAM dendrimers with modification of phenylboronic acid (Dan-B-PAMAM) were designed and their interaction with bacteria was investigated. Visible aggregates were obtained. Their size depended on the number of terminal phenylboronic acids on dendrimers, which turned out to be an indicator of increased sensitivity. The main driving force behind the aggregation was the interaction between phenylboronic acids and glycolipids on the bacterial surface, not only the electrostatic interactions of the amino groups of the dendrimers. Phenylboronic acids, after recognition, selectively influenced the viability of E. coli. This test could be a potential method for convenient and sensitive detection and killing of bacteria. Description of the experiments, results and their discussion does not raise any major objections.

The paper could be published in the present form.

Author Response

(The authors gave the same response as above.)
